# Measurement Approach for the Pose of Flanges in Cabin Assemblies through Distributed Vision

**DOI:** 10.3390/s24144484

**Published:** 2024-07-11

**Authors:** Xiaojie Ma, Jieyu Zhang, Tianchao Miao, Fawen Xie, Zhongqiu Geng

**Affiliations:** 1School of Mechanical Engineering, Henan Institute of Technology, Xinxiang 453003, China; maxiaojie@hait.edu.cn (X.M.); miao.csts@hait.edu.cn (T.M.); gzq040520@stu.hait.edu.cn (Z.G.); 2SHIEN-MING WU School of Intelligent Engineering, South China University of Technology, Guangzhou 511442, China; fwingtks@163.com

**Keywords:** cabin, automatic assembly, distributed vision, pose measurement, registration, precision measurement

## Abstract

The relative rotation angle between two cabins should be automatically and precisely obtained during automated assembly processes for spacecraft and aircraft. This paper introduces a method to solve this problem based on distributed vision, where two groups of cameras are employed to take images of mating features, such as dowel pins and holes, in oblique directions. Then, the relative rotation between the mating flanges of two cabins is calculated. The key point is the registration of the distributed cameras; thus, a simple and practical registration process is designed. It is assumed that there are rigid and scaling transformations among the world coordinate systems (WCS) of each camera. Therefore, the rigid-correct and scaling-correct matrices are adopted to register the cameras. An auxiliary registration device with known features is designed and moved in the cameras’ field of view (FOV) to obtain the matrix parameters so that each camera acquires traces of every feature. The parameters can be solved using a genetic algorithm based on the known geometric relationships between the trajectories on the registration devices. This paper designs a prototype to verify the method. The precision reaches 0.02° in the measuring space of 340 mm.

## 1. Introduction

When assembling spacecraft and aircraft, accurately measuring the relative rotation angle between two cabins is vital for precise docking [1].

This process significantly influences the final product’s performance. For example, relative positional deviations of mating features on the mating surface will cause collisions during the assembly process, while deviations in the pose of the mating feature axes will lead to jamming or wedging during assembly. Hence, developing an automated and precise technique is essential for replacing conventional manual approaches in cabin assembly. A general method for these requirements can be summarized as “laser tracker system (LTS) + 3-degree of freedom (DOF) coordinators”, refs. [2,3,4], in which the three-DOF coordinators are adopted as a group to sustain the cabin and adjust their pose based on the measurement result from the LTS.

However, the mating feature poses cannot be acquired directly through this approach, which introduces operation and transfer errors that may overwhelm the true results when operating the LTS probe [5]. In some studies [6,7,8], pose estimation of mating features is achieved by measuring the position of markers sprayed on the surface through vision approaches, but this has the same problem as the LTS. Therefore, when high precision is required, it is imperative to directly measure the pose of mating features to reduce errors.

To avoid errors introduced by measuring probes or marks, 2D structured light can be introduced to acquire the external point clouds of the cabins directly, from which the pose can be solved through the iterative closest point algorithm (ICP) [9]. For example, Hu et al. [10] combined a projector and binocular cameras to form a structured light system for rapid estimation of the 6D pose of objects, achieving a measurement accuracy of 0.27 mm/0.38°. Wang et al. [11] addressed the assembly issues of large-scale cabins by employing a surface structured light sensor to acquire the point cloud of the cabins for docking. Subsequently, both the iterative closest point (ICP) and random sample consensus (RANSAC) are employed to determine the relative poses between cabins. However, this method exhibited poor measurement accuracy for the rotation angle of the parts around its own axis. Liu et al. [12] positioned the structured light system at the end of a rotatable three-axis platform, thereby expanding the device’s measurement range for assessing the poses of large cabin components, and the measurement accuracy reached 0.2 mm/0.2°. However, such surface structured light systems necessitate projectors to sequentially project distinct stripe patterns in a temporal sequence [13], hindering the real-time measurement of cabin poses. Furthermore, the method exhibits inadequate measurement accuracy for the rotation angle of cabins around their axis. Using time-of-flight (TOF) technology enables the real-time capture of three-dimensional point clouds of the measured object [14,15]. However, the precision of this method is poor, making it difficult to achieve accurate measurements of object poses during assembly.

Many studies have been dedicated to improving the measurement precision of cabin poses during assembly processes and addressing real-time measurement problems. Raytheon designed special ring fixtures to hold missile cabins and ensure their axes were coincident [16]. Liu et al. design a four-degree-of-freedom adjustable U-shaped support capable of accommodating elongated cylindrical cabins and adjusting their pose [17]. Zhang et al. [18] adopted a laser scanner to acquire surface point cloud data and solve the axis pose. Robots were then employed to align the cabins. Hao et al. [19] ensured precise alignment of the axes of two cabin segments during assembly by incorporating force-compliant techniques. These approaches have yet to address the crucial aspect of measuring relative rotation. This measurement is essential for the final step of cabin assembly, involving cabin rotation and subsequent joining.

Due to the axial symmetry of cabins, it is necessary to determine the relative rotation angle between two cabin segments based on certain mating features on the docking end faces [20]. For example, Hao [21] designed a method to measure rotations based on two laser-ranging sensors mounted back-to-back on a rotation device to tackle this problem. They calculated the depth information of the annular region on the mating surface and the relative rotation angle between the cabins. However, the rotation device was located between two mating surfaces, hindering the subsequent assembly process. Li et al. [22] proposed a coaxial alignment method for large aircraft assembly based on distributed monocular vision. Cameras were mounted coaxially with a hole in one part, and pictures were taken through the hole of another part. Then, an iterative reweighted particle swarm optimization (IR-PSO) algorithm was employed to calculate the relative poses of the parts. However, this method is limited to cabin assembly as it is only suitable for hole-to-hole assembly, indicating sensors should be mounted inside the part to be assembled. Liu et al. [23] measured the pose of parts in a micro-assembly using three cameras arranged perpendicularly. A six-DOF manipulator was employed to finish the posture alignment. This technology provides a solution for measurements in a micro-assembly. However, it is not suitable for large parts in practical engineering environments that always have unstructured backgrounds and illumination.

In response to this challenge, this paper introduces a measurement method based on distributed vision. The approach employs a set of monocular cameras strategically positioned along the mating surface of the cabin segments to individually capture mating features on the end face. Subsequently, the positions of these features are calculated within their respective measurement fields. A measurement field registration method is then proposed, where the mating features are transformed into a unified coordinate system, facilitating computations of the relative poses between two cabin segments. There is only one specific predetermined feature presented on the mating surface within the fields of view (FOVs) for each camera. Thus, the approach allows more precise position information extraction, effectively resolving the inherent contradiction between the camera FOV and precision. Consequently, it enables accurate control of the attitude adjustment mechanism to fine-tune the cabin’s angular error. Finally, experimental validation using a prototype shows that the measurement accuracy of the proposed approach surpasses 0.05 mm for a cabin with a diameter of 340 mm. Leveraging the proposed registration method makes it feasible to unify the coordinate systems of multiple cameras with non-overlapping FOVs. As a result, there is substantial potential for widespread application in vision-based assembly systems.

The content of this paper is organized as follows. In Section 2, a distributed visual measurement method for cabin rotation angles is designed to precisely measure angular errors between two cabin segments without relying on measurement targets. Section 3 presents the registration method for distributed cameras to unify the global coordinate system of the camera array. Section 4 summarizes the proposed method and its applications in practical industrial assembly.

## 2. Principles and Methods

### 2.1. Flange Pose Measurement Method

As illustrated in Figure 1, the poses of cylindrical components with mating features, such as pins or holes on the end face, can be described in the global coordinate system (GCS) using six pose parameters, $T=\left(x_{0},y_{0},z_{0},\alpha ,\beta ,\gamma \right)$. These parameters include the displacement deviations of the flange center in the three coordinate axis directions, and the assembly rotation angle, α, about $Z_{0}$, the deflection angle, β, about $Y_{0}$, and the pitch angle, γ, about $X_{0}$.

In previous studies, the pose of the cylindrical component axis, represented by five parameters $\left(x_{0},y_{0},\alpha ,\beta ,\gamma \right)$ in T, was ensured through V blocks or specialized circular fixtures [24]. Alternatively, precise measurements are achieved through structured light scanning methods [18]. However, these methods still prove ineffective in accurately measuring the assembly rotation angle, α, for cylindrical components with flanges.

Positioning the camera directly in front of the segment end face for spacecraft mating is difficult. To solve this problem, we introduce a method that tilts the camera inward along the circumferential edge of the docking end face to capture images of the mating features, as illustrated in Figure 2. This approach employs two sets of industrial cameras, numbered and arranged radially to capture pictures of the mating surface of two cylindrical components, A and B. This configuration forms a distributed monocular system, with the FOV for Camera Group $A_{i}$ encompassing the mating features $P_{Ai}$(i=1,⋯,N) on the assembly face of cylindrical component A, and likewise for cylindrical component B.

For vision measurement, camera calibration is essential. Calibration provides the necessary camera parameters, which include both intrinsic and extrinsic parameters. The intrinsic parameters encompass information such as the camera’s focal length, distortion, and pixel parameters, while the extrinsic parameters refer to the camera’s pose relative to the imaging plane [25]. As shown in Figure 3, the extrinsic parameters indicate the transformation relationship between the coordinate system defined by a specific calibration board center and the camera pose. Here, this calibration board coordinate system is referred to as the camera’s world coordinate system (WCS). However, as each camera has its own distinct WCS corresponding to its measurement field, it is referred to as the camera world coordinate system (CWCS) for distinction. Therefore, calibration establishes the mapping relationship between the camera pixel coordinate system (PCS) and the CWCS.

Then, the camera is calibrated by attaching the calibration board to the plane of each mating feature, thereby establishing the CWCS, as shown in Figure 3. It is clearly that the XOY planes of each camera’s CWCS are parallel to each other. Furthermore, the positions of respective docking features can be obtained through conventional machine vision measurements in their corresponding CWCS. However, the relative positions and the relative rotational positions around the Z axis cannot be determined by individual calibrations. Therefore, employing a suitable global calibration method to register the various measurement fields is still necessary. This registration process transforms the positions of each mating feature into a unified GCS for comparative analysis. Subsequently, the pose deviations of individual features on the two mating surfaces can be determined.

Optical devices such as encoders and laser rangefinders are employed during operations to ensure that the support mechanism moves the mating surfaces of the two cabins to a pre-calibrated imaging plane. This guarantees a fixed value for $Z_{0}$ during each measurement. Camera groups $A_{i}$ and $B_{i}$ capture images of the corresponding pairs of mating features on the end face, extracting the coordinates of their center points in their respective CWCSs, denoted as PAiAi and PBiBi. Therefore, the CWCS of each camera can be illustrated as in Figure 4.

CWCS transformation can be accomplished through a homogeneous transformation matrix, H, for cameras capturing mating feature pairs. For a given mating feature pair, position measurements by camera groups A and B yield the respective coordinates PAiAi and PBiBi in their CWCS. Their relationship can be expressed as follows:(1)PAiBi=HBiAiPBiBi

Utilizing Equation (1), the position measured by Camera Group B can be transformed into the CWCS of Camera Group A. If the two cylindrical components are precisely aligned coaxially through fixtures or adjustment mechanisms, the angular error between the corresponding mating features of the cylindrical components can be calculated as follows:(2)Δα=2arcsinPA0Ai−PA0Bi2R
where R is the radius of the center point of the mating feature. Therefore, calibration should be performed before measurements in two aspects: individually calibrating each camera to accurately determine the positions of mating features in their respective CWCS, and globally aligning all cameras to obtain the registration matrix, H, between camera groups.

Each camera captures images of the mating features within its FOV during the measurements. The obtained images are corrected for distortion and perspective-induced deformations using calibration information. Subsequently, the center positions of the mating features in their respective CWCS are extracted from the corrected images. Following this, the registration matrix, H, transforms these centers into a unified GCS. The positional errors between corresponding mating features are then calculated.

### 2.2. Acquiring the Positions of Flange Mating Features in CWCS

As illustrated in Figure 5, the subpixel position (u,v) of the hole in the image acquired by a camera can be extracted through subpixel image processing. Based on the intrinsic and extrinsic matrices obtained from the camera calibrations, the subpixel position is reprojected to determine its corresponding position, PAiAi(xW,yW) or PBiBi(xW,yW), in its respective CWCS [25].

In Figure 5, step (a) involves a Gamma transformation to enhance the dark regions of the image for improved contrast in the measurement features, facilitating threshold segmentation [26]. Then, median filtering is utilized to smooth the image and reduce the adverse effects of texture on threshold segmentation [27]. Step (b) employs threshold segmentation to extract the processing region of interest (ROI) where the hole edges are located, helping determine the hole’s edge position. Step (c) performs an intersection operation between the ROI and image after the Gamma transformation to determine the edge extraction region. Finally, the Canny operator is applied to extract hole edges, followed by elliptical fitting to determine the position of the hole center in the image [28]. This process yields a hole center position in its CWCS.

### 2.3. Registration of Distributed Cameras of Mating Surface

When docking cabins, calculating the positional errors between mating features, such as hole-to-hole and pin-to-hole connections, is often necessary to determine the relative angular displacement between two flanges. Therefore, camera registration for relative mating surfaces is required. This requires the transformation of the CWCS for each camera to a common coordinate system through the matrix HBiAi, as depicted in Figure 4 and Equation (1).

A registration device was devised, as illustrated in Figure 6. The approach utilized a precise registration mechanism to align and calibrate two cameras. In Figure 6a, the calibration planes on both sides of the registration device feature a pair of coaxial calibration holes. The distance between the two calibration planes equals the distance between the two flanges while imaging, simulating the camera-to-hole distance during actual hole measurements.

During registration, the two inner surfaces of the registration device must be aligned with the planes where the two mating surfaces are located during imaging. This ensures that the intrinsic and extrinsic parameters do not change during registration. Each camera simultaneously captures images of the left and right holes on the registration device, providing a pair of center coordinates as the registration point sets PAk(xAk,yAk) and PBk(xBk,yBk), k=1,⋯,N. Sliding the registration device along the outer wall of the cabin segment provides N sets of registration point pairs.

As mentioned earlier, the CWCS of the two cameras can be registered using the transformation matrix HBA, as given in Equation (1). To simplify the solution, performing a pre-transformation on the registration point set PBk, denoted as P^Bk, such that the centroids of the two point sets coincide, is necessary. The transformation formula is given by the following:(3)P^Bk=HBprePBk
where the point set HBpre is the pre-transformation matrix,
(4)HBpre=10xBA01yBA001

$\left(x_{BA},y_{BA}\right)$ is the relative position of the centroid for point set A with respect to the centroid of point set B. Thus, for any point in the CWCS of Camera B, the corresponding point in the CWCS of Camera A is obtained through the transformation matrix, HBA, as follows:(5)P^Ak=HBAtrans⋅P^Bk
where the transformation matrix, HBAtrans, is one possible form of HBA that denotes the translational transformation of trajectory B to a state coinciding with trajectory A. The transformation matrix is represented as follows:(6)HBAtrans(θ,xt,yt)=cosθ−sinθxtsinθcosθyt001

Since trajectory A and trajectory B should coincide after registration, trajectory B, transformed by Equations (3) and (5), should overlap with trajectory A. Therefore, the sum of the algebraic distances between corresponding points of trajectory A and the transformed trajectory B can be calculated and used as the objective function for optimization. Therefore, the registration process can be represented as an optimization procedure:(7)argminθ,xt,yt∑kNPAk−HBAtrans⋅P^Bk

In Equation (7), the coordinates of the point set PAk in trajectory A and the coordinates of the transformed point set P^Ak in trajectory B are known. Thus, we apply a genetic algorithm to optimize Equation (7) to obtain the parameters $\left(\theta ,x_{t},y_{t}\right)$ in the transformation matrix HBAtrans. Consequently, all points in Camera B can be transformed into Camera Group A through the following:(8)P^Ak=HBAtransHBprePBk

As a result, the registration transformation matrix from Camera B to Camera A is denoted as follows:(9)HBA=HBAtransHBpre

Owing to variations in camera poses and additional errors stemming from scaling, these discrepancies cannot always be rectified through translational transformation, HBAtrans. An enhancement is introduced to Equation (5) to solve this problem as a scaling correction matrix to further refine the transformed points, i.e.,
(10)P^Ak=HBAscaleHBAtransP^Bk

The form of the scaling correction matrix is as follows:(11)$$H_{scale}(f_{x},f_{y})=\left[\begin{array}{ccc} f_{x} & 0 & 0\\ 0 & f_{y} & 0\\ 0 & 0 & 1 \end{array}\right]$$

The parameters $\left(f_{x},f_{y}\right)$ represent dimensionless correction coefficients along the X and Y axes, respectively. Thus, registration transformation from Cameras B to A can be expressed as follows:(12)PAk=HBAscaleHBAtransHBprePBk

Equation (7) can then be written as follows:(13)argminθ,xt,yt,fx,fy∑k=1MPA−HBAscaleHBAtransP^Bk

Taking scaling correction into consideration, the registration transformation matrix from Camera B to Camera A can be written as follows:(14)HBA=HBAtransHBpre

## 3. Experiment and Result

An experimental system was constructed to validate the accuracy of the proposed method, as illustrated in Figure 7. In the setup, holes are positioned on the flanges of the cabin and target cabin segments. Simultaneously, the alignment of the two cylindrical components is ensured through a V block to maintain coaxiality.

First, the two cameras are individually calibrated, and their respective CWCSs are established. Then, the registration device is positioned to slide between the two cylindrical components. Both cameras continuously capture images of their respective registration holes on the devices. The trajectory is plotted in Figure 8a.

The trajectory points captured by Camera A are denoted as PAk, and those captured by Camera B are denoted as PBk. Subsequently, PBk undergoes a pre-transformation using Equation (3), resulting in P^Bk. Substituting PAk and P^Bk into Equations (7) and (13), respectively, with initial values set to (0,0,0) for parameters $\left(\theta ,x_{t},y_{t}\right)$ and (0,0,0,1,1) for $\left(\theta ,x_{t},y_{t},f_{x},f_{y}\right)$, the pose and scaling parameters in Table 1 were obtained using genetic algorithms implemented in the Matlab optimization toolbox.

Thus, trajectory B after being transformed in accordance with the parameters acquired in Table 1 and Table 2 is shown in Figure 8b. In order to evaluate the effectiveness of the two registration equations, i.e., Equations (8) and (12), the residual distribution curve based on the distance between corresponding points after the transformation is shown in Figure 9.

As shown in Figure 9a, a residual curve without scale correction indicates significant errors. Therefore, in addition to differences in rigid body transformation, there is also a difference in scaling between the true coordinate systems of the two cameras. Therein, the results obtained via this transformation are not ideal. Comparatively, the registration errors with scale correction shown in Figure 9b decreased by an order of magnitude.

Therefore, it can be concluded that the registration relationship between the two cameras should include both translation and scaling transformations, implying that the registration process described in Equations (10)–(14) achieves higher precision.

As shown in Figure 10, to verify the reliability and accuracy of the proposed method, this study designed an experiment focusing on the mating features in an actual cabin assembly system.

In this experiment, after successfully docking two simulated segments with diameters of 340 mm using high-precision pin locating and fixing the relative rotation angle of the segments, the two mated surfaces were separated and moved to the photographing position to validate the effectiveness of the proposed method. The measured positions of the holes were obtained after removing the locating pins. As the projections of the holes in the global coordinates should coincide, the hole positions measured by Camera B were transformed using Equations (12) and compared with the hole positions from Camera A. The relative positional deviations were calculated, and Equation (2) provided the relative rotation angle of the two segments. This process was repeated five times to evaluate the accuracy of the proposed method. The results are presented in Table 2.

The results in Table 2 indicate that after image registration, the theoretical angular error, Δα, calculated using Equation (2) is expected to be 0°. In practice, the maximum observed measurement error is 0.016°, and all calculated relative angular values, Δα, between the flanges are within 0.02°. Considering the potential errors in image processing and inherent mechanical inaccuracies in the assembly platform, the proposed registration method demonstrates high accuracy.

## 4. Conclusions

This paper introduces a precision measurement method for the rotational angles and poses of flanges on cylindrical components meant for aerospace module units to address common alignment and assembly challenges in the final assembly process. The proposed method leverages distributed vision to allow for automatic and efficient precision measurements of key rotational angles and poses during the docking process of spacecraft sections without needing measurement targets or probes. This enables closed-loop adjustments to servo systems to precisely align rotational angles and poses during docking. The applicability of the proposed method extends beyond the docking of cylindrical sections and covers scenarios involving non-circular end-face dockings and multiple camera setups. The measurement and camera registration processes remain analogous in such scenarios.

This study presents a camera registration model incorporating rigid body transformation matrices, $H_{trans}$, scaling correction matrices, $H_{scale}$, and a feature-based registration method based on experimental error quantification in the distributed camera registration process. This method ensures the uniformity of the WCS for camera pairs that capture mating features, facilitating the measurement of relative rotational angles and poses between cabins.

Finally, a prototype was developed to validate and confirm the proposed method’s accuracy. Practical measurements revealed a relative rotational angle error of less than 0.02° between the cabins, demonstrating high precision. Unlike traditional approaches relying on laser trackers, the proposed method allows for the direct measurement of mating features, eliminating the requirement of target probes. As a result, the device configuration is flexible and can easily integrate with closed-loop control systems, offering a convenient solution for the automated enhancement of existing assembly production lines.

## Figures and Tables

**Figure 1 sensors-24-04484-f001:**
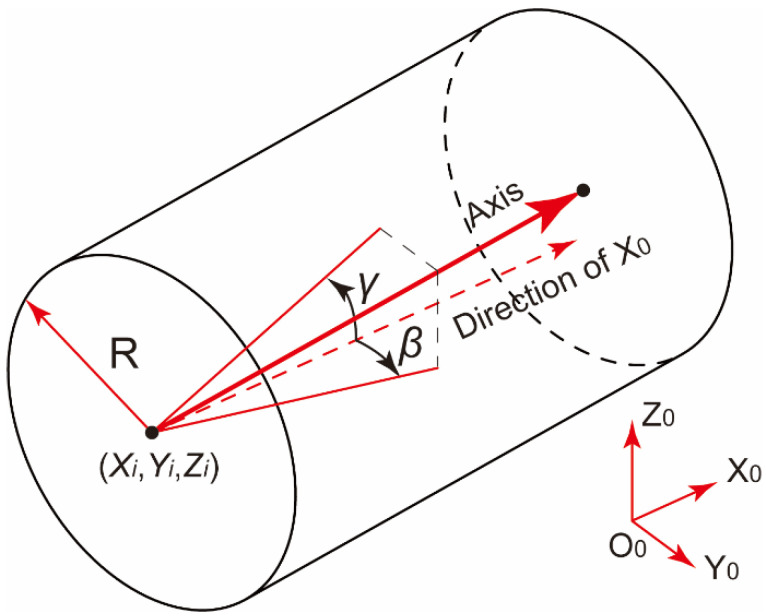
Schematic diagram of pose parameters for cylindrical components.

**Figure 2 sensors-24-04484-f002:**
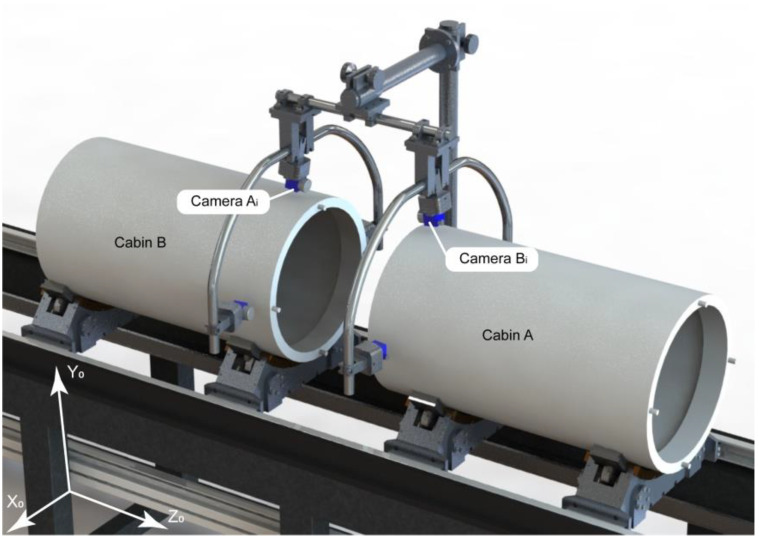
Schematic diagram of proposed measuring device for mating aerospace components.

**Figure 3 sensors-24-04484-f003:**
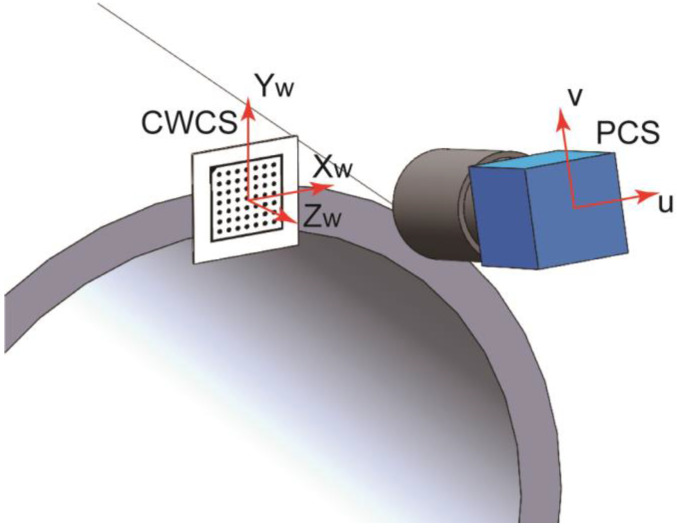
Illustration of the relationship between the CWCS and PCS.

**Figure 4 sensors-24-04484-f004:**
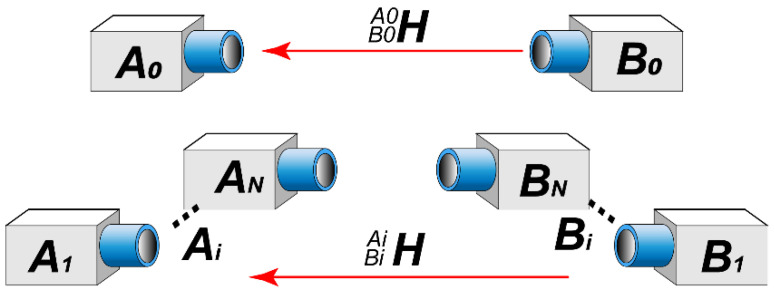
Illustration of the relationship between the CWCSs of each camera.

**Figure 5 sensors-24-04484-f005:**
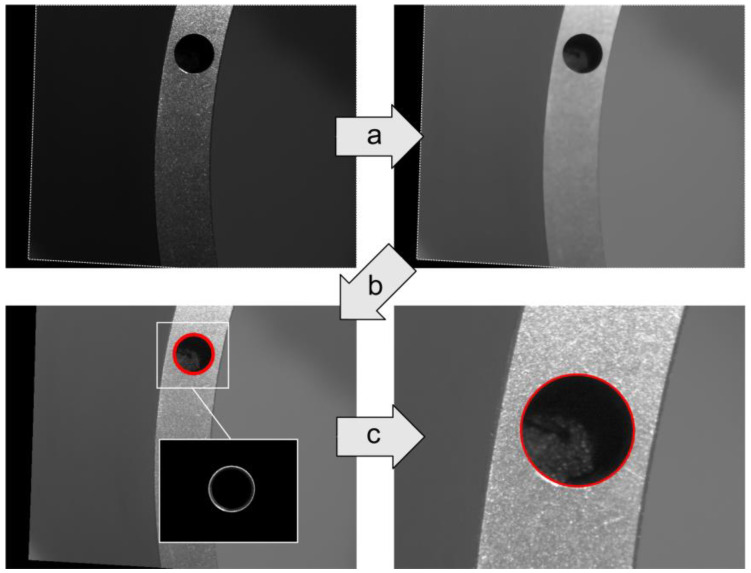
The subpixel extraction process for a hole.

**Figure 6 sensors-24-04484-f006:**
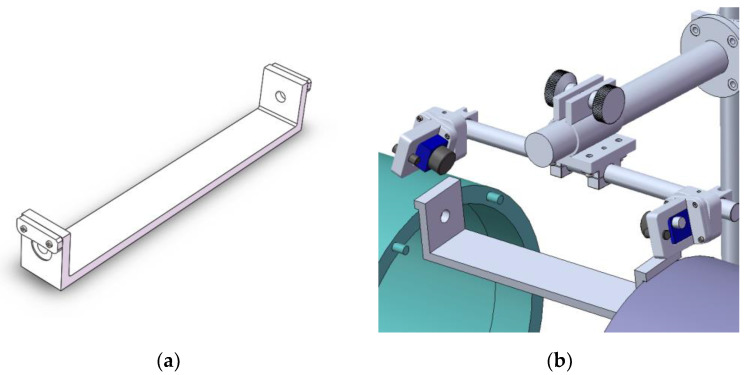
Camera registration method using the auxiliary registration device. (**a**) Registration device; (**b**) working schematic of the device.

**Figure 7 sensors-24-04484-f007:**
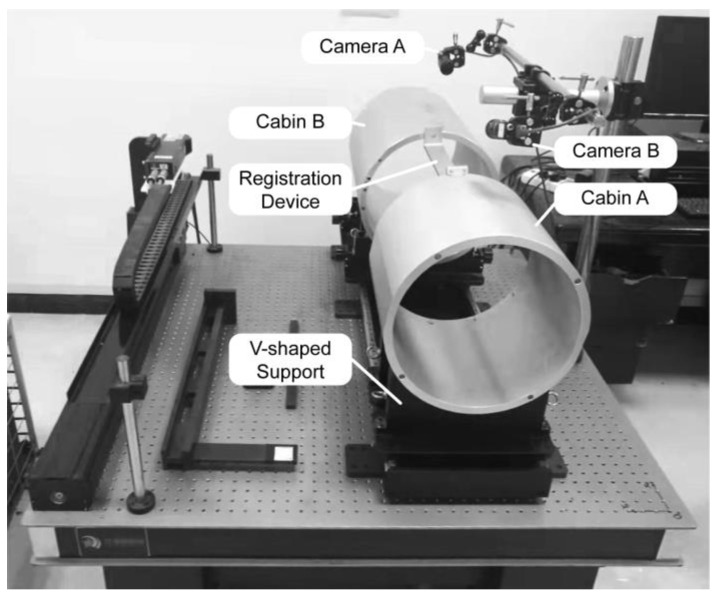
Experimental setup for camera registration on opposite surfaces.

**Figure 8 sensors-24-04484-f008:**
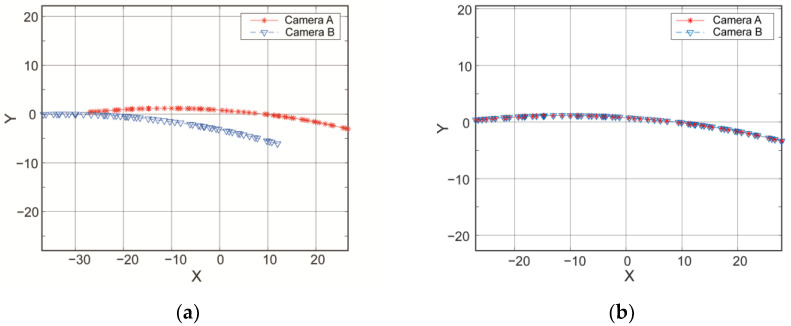
Comparison of coordinates for the holes before and after registration. (**a**) Coordinates of the holes before registration. (**b**) Coordinates of the holes after registration.

**Figure 9 sensors-24-04484-f009:**
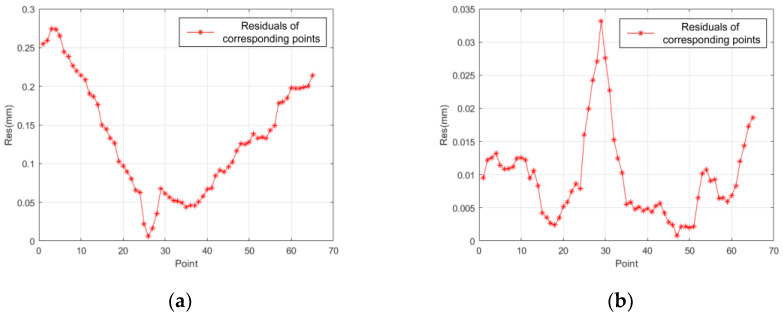
Comparison of residuals before and after applying the scale correction. (**a**) Residual curve without scale correction. (**b**) Residual curve with scale correction.

**Figure 10 sensors-24-04484-f010:**
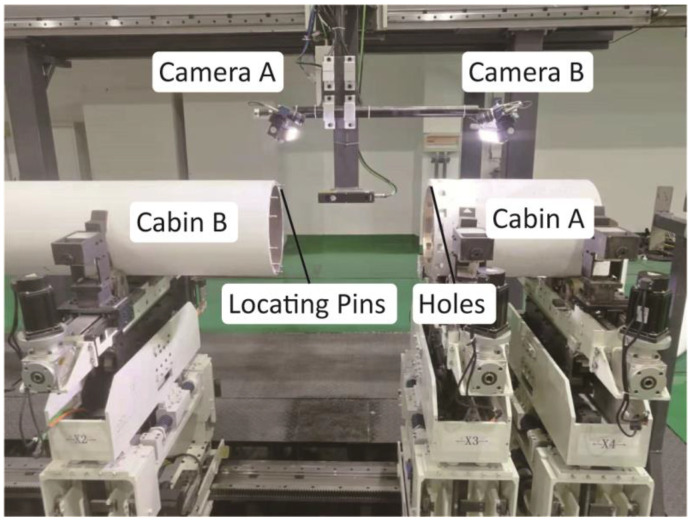
Experiment in an actual cabin assembly system.

**Table 1 sensors-24-04484-t001:** Values of parameters for Equations (7) and (13).

Parameter	θ (rad)	$x_{t}$ (mm)	$y_{t}$ (mm)	$f_{x}$	$f_{y}$
Equation (7)	0.039	−1.199	1.115	-	-
Equation (13)	0.038	−1.083	1.720	1.009	1.001

**Table 2 sensors-24-04484-t002:** Registration error statistics.

No.	Hole Coordinate of Camera A		Hole Coordinate of Camera B after Transformation		Position Deviation (mm)	Angle Deviation (°)
Coordinate	X (mm)	Y (mm)	X (mm)	Y (mm)		-
1	21.083	−21.462	21.127	−21.471	0.045	0.016
2	−6.832	−19.711	−6.796	−19.732	0.042	0.015
3	−11.819	−19.567	−11.801	−19.551	0.024	0.008
4	−25.205	−20.882	−25.215	−20.867	0.018	0.006
5	−34.431	−22.397	−34.456	−22.405	0.026	0.009
Maximum deviation	-				0.045	0.016

## Data Availability

Data are contained within the article.

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
