# Peer review of "Measurement Approach for the Pose of Flanges in Cabin Assemblies through Distributed Vision"

_sensors, 2024, doi:10.3390/s24144484_

Round 1
Reviewer 1 Report
Comments and Suggestions for Authors
This article is about method to solve this problem based on distributed vision, where two groups of cameras are employed to take images of mating features, such as dowel pins and holes, in oblique directions.
Basically, there is nothing fundamental to criticize about the article. It is written according to standards and has all the requisites of a scientific article. I like the use of mathematics and its explanation within the article. The number of cited sources also corresponds to this issue.
Perhaps only tables 1 and 2 will be useless. I would either connect them in some way or replace them with a vertical table.
Author Response
Dear Reviewer,
Thank you for your careful review and constructive feedback on our manuscript titled Measurement Approach for the Pose of Flanges in Cabin Assemblies through Distributed Vision. We appreciate the time and effort you have dedicated to evaluating our work.
Regarding tables 1 and 2, we’ve connected them together as a vertical table and enhanced the description of the data acquisition process by providing additional details.
Thank you once again for your insightful feedback and guidance throughout this process. We look forward to your further input.
Best regards

Reviewer 2 Report
Comments and Suggestions for Authors
The article refers to the registration of distributed cameras used in measuring rotation angles between two cylindrical components for their coupling during assembly. Both components include holes or pins in their respective flanges, which were not used since the pins were removed during the experiments.
I have serious doubts about the suitability of the assembly method, since the main problem is always the mating pairs considering the tolerances, since normally the assembly could end with jamming or wedging, and the presented method was not verified for the assembly process.
Apart from that, there are some points that make the article difficult to read, for example:
The homogeneous transformation matrix for the use of both cameras is explained. However, it is not clear how the perspective angle is compensated due to camera positions that is typically achieved with homography. An explanation is required.
The authors state in section 3, page 9 that " The coordinates of the points were substituted into Equation (7) and (13), and their values for the parameters in the transformation matrix were solved using a genetic algorithm." However, there is no explanation on how the GA works, its restrictions, operations, etc.
Figure 4 needs to be explained in terms of how the image processing is done: a)-b)-c)-d) and then b) again it's confusing. So, under what conditions is step e) carried out?
Author Response
Dear Reviewer,
Thank you for your careful review and constructive feedback on our manuscript titled Measurement Approach for the Pose of Flanges in Cabin Assemblies through Distributed Vision. We appreciate the time and effort you have dedicated to evaluating our work.
The details of the manuscript revisions and our responses are attached in the attachment. Please see the attachment.
Thank you once again for your insightful feedback and guidance throughout this process. We look forward to your further input.
Best regards

Round 2
Reviewer 2 Report
Comments and Suggestions for Authors
Authors improved the camera calibration explanation including a new figure 3.
They also described better figure 4(now figure 5).
Equation 7 is better explained including Table 1 and mentioning the used toolbox.